# Comprehensive Evaluation of 17 Qualities of 84 Types of Rice Based on Principal Component Analysis

**DOI:** 10.3390/foods10112883

**Published:** 2021-11-22

**Authors:** Shijie Shi, Enting Wang, Chengxuan Li, Hui Zhou, Mingli Cai, Cougui Cao, Yang Jiang

**Affiliations:** 1College of Plant Science and Technology, Huazhong Agricultural University, Wuhan 430070, China; shishijie@webmail.hzau.edu.cn (S.S.); etwang@webmail.hzau.edu.cn (E.W.); muyan20210601@163.com (C.L.); huizhou@webmail.hzau.edu.cn (H.Z.); caimingli@mail.hzau.edu.cn (M.C.); ccgui@mail.hzau.edu.cn (C.C.); 2Hubei Collaborative Innovation Center for Grain Industry, Yangtze University, Jingzhou 434025, China

**Keywords:** rice, appearance quality, milling quality, cooking and eating quality, principal component analysis

## Abstract

Rice quality is a complex indicator, and people are paying more and more attention to the quality of rice. Therefore, we used seven rice varieties for twelve nitrogen fertilizer treatments and obtained eighty-four rice types with seventeen qualities. It was found that 17 quality traits had different coefficients of variation. Among them, the coefficient of variation of chalkiness and protein content was the largest, 44.60% and 17.89% respectively. The cluster analysis method was used to define four categories of different rice qualities. The principal component analysis method was used to comprehensively evaluate 17 qualities of 84 rice. It was found that rice quality was better under low nitrogen conditions, Huanghuazhan and Lvyinzhan were easier to obtain better comprehensive rice quality during cultivation. Future rice research should focus on reducing protein content and increasing peak viscosity.

## 1. Introduction

Rice is the staple food of two-thirds of the world, and the production of rice is very important [1]. China is the most important rice producer in the world, and about half of the people feed on rice as their staple food [2]. Different from wheat, corn, and other crops, rice is usually converted into polished rice after being hulled and milled, which can then be eaten by people [3]. With the increase of living standards, people pay more and more attention to the quality of rice [4]. The quality of rice includes the appearance-, milling-, cooking-, and eating quality of rice. Appearance quality refers to the length, width, and chalkiness of rice. In terms of grain type, people in some countries like long rice grains. Long-grain rice has soft grains and has a higher elongation rate after being boiled [5]. Chalkiness affects the overall appearance of rice, and people tend to prefer rice without chalkiness [6]. Milling quality usually refers to the proportion of rice that has been dehulled and milled into polished rice. After rice with poor processing quality is milled, the proportion of rice milled into polished rice decreases, indicating that there is more waste in the processing process [7]. Cooking and eating quality refers to the taste of rice, high-taste rice has a brighter appearance and softer taste [8]. In the past, it was often only possible to taste rice directly in order to correctly evaluate the taste quality of the rice [9]. However, this method is often cumbersome, and the results are inaccurate due to differences in age, region, and preference among different people. In addition to direct tasting, the eating quality of rice can also be measured indirectly through RVA. High eating quality rice tends to have a higher peak viscosity and a lower gelatinization temperature [10]. A recent rice taste analyzer can evaluate the taste value of rice. Rice with high cooking- and eating quality tends to have a higher taste value [11].

Nitrogen fertilizer is one of the most important factors affecting the cooking and eating quality of rice. Appropriate nitrogen increases the yield of rice, but it will affect the quality of rice. Nitrogen fertilizer improves the milling quality of rice [12], reduces the taste of rice, and ultimately leads to poor-tasting rice [13]. Recently, a comprehensive evaluation analysis on foxtail millet quality indicated that future research on foxtail millet should focus on increasing protein content and amino acids [14]. Some key indicators can be used to evaluate the high-temperature tolerance of different rice varieties, to select the best varieties [15]. However, there is not a lot of research on rice quality, and research is needed to guide rice production through a comprehensive analysis of rice quality. For a long time, the effect of nitrogen on rice quality was often concentrated in a single quality study. There are often few rice varieties used, the quality evaluation of rice is not representative, and there are no comprehensive evaluations of rice quality.

Therefore, we used principal component analysis to evaluate rice quality comprehensively; it helped researchers determine good rice varieties and cultivation conditions. In this study, seven widely grown rice varieties, including three japonica rice varieties and four indica rice varieties, were subjected to twelve nitrogen fertilizer treatments in the field test, resulting in eighty-four different types of seventeen rice qualities. The quality of rice (appearance-, milling-, cooking-, and eating quality) for a comprehensive evaluation, to provide their suggestions for high-quality rice production.

## 2. Materials and Methods

In 2020, the seven rice varieties (Tianyuanxiangjing, Yongyou7850, Yongyou4949, Huanghuazhan, Lvyinzhan, Jiafengyou II, Meixiangzhan II) with different grain shapes and different taste values were used for field trials. Indica rice included Huanghuazhan, Meixiangzhan, Lvyinzhan, and Jiafengyou II. Japonica rice included Yongyou7850, Yongyou4949 and Tianyuanxiangjing. The field experiment was conducted at a research farm of Jianli County, Hubei Province, China (30°5′ N, 112°56′ E) during the rice-growing season. Plots were arranged in a randomized block pattern with three replicates. Seeds were sown on 1 June 2020, and seedlings were transplanted on 5 July 2020. Rice seedlings were planted in one seeding per hole and transplanted at an interval of 30 × 13 cm. Superphosphate (50 kg ha^−1^) and potassium chloride (100 kg ha^−1^) were fertilized once before transplanting. 12 nitrogen fertilizer treatments including 0 kg N ha^−1^, 25 kg N ha^−1^, 50 kg N ha^−1^, 75 kg N ha^−1^, 100 kg N ha^−1^, 125 kg N ha^−1^, 150 kg N ha^−1^, 175 kg N ha^−1^, 200 kg N ha^−1^, 250 kg N ha^−1^, 300 kg N ha^−1^, and 350 kg N ha^−1^, the total amount of N were applied with the ratio of 5:3 at pre-transplanting and green turning. We used a rice polisher (Satake, Tokyo, Japan) for further experimental analysis.

### 2.1. Appearance Quality

A rice appearance quality analyzer (SC-E, Hangzhou Wanshen Test Technology Corporation, Hangzhou, China) was used to analyze the appearance quality of milled rice [16], including length, width, and chalkiness. Grains containing more than 20% white belly, white center, white back, or a combination of these are considered chalk particles.

### 2.2. Milling Quality

Husker machine (Satake, Tokyo, Japan) was used for husking of 25 g paddy samples. Brown rice percentage was calculated by dividing the weight of brown rice by 25 g and multiplying by 100. Then grind the brown rice into milled rice with Kett mill (Kett, Tokyo, Japan), milled rice percentage was calculated by dividing the weight of milled rice by 25 g and multiplying by 100. Calculated the brown rice percentage and milled rice percentage of the rice. Head rice refers to the kernels that remain at three-fourths or more of their normal length [17].

### 2.3. Cooking and Eating Quality

#### 2.3.1. Protein and Amylose Contents

The rice samples were pulverized (Foss, Hilleroed, Denmark) and passed through a 100-mesh aperture. The N concentration of milled rice was determined using an Elemental analyzer (Elementar, Langenselbold, Germany), then converted into protein content using a conversion factor (5.95) [18]. The amylose content was determined by the iodometric method [19]. An Epoch Microplate Spectrophotometer (BioTek, Winooski, VT, USA) was used to measure the color at 620 nm. The amylose content values were calculated from a standard curve established using mixture solutions of amylose and amylopectin.

#### 2.3.2. RVA Profiles

A rapid viscosity analyzer RVA (Newport, Warriwood, Sydney, NSW, Australia) was used to determine the RVA profiles of rice. According to AACC (American Association of Cereal Chemists, Saint Paul, MN, UAS) procedures and Windows Thermal Cycle (TCW3) for data processing and data analysis [20]. About 3 g of rice flour was mixed with 25 mL of water and then put on an aluminum can. The RVA program was first heated at 50 °C for one minute, then heated to 95 °C in 3.75 min, and then heated at 95 °C for 2.5 min. It was then cooled to 50 °C in 3.75 min and held for 1.4 min. Head rice samples (2 g) were cooked in 20 mL distilled water for 20 min in a boiling water bath. The contents were drained, and cooked samples were then weighed accurately and the water uptake ratio was calculated.

#### 2.3.3. Taste Values

The rice taste analyzer (Satake, Hiroshima, Japan) was used to determine the taste value of rice. The rice taste analyzer converted various physical and chemical properties of rice into taste values to determine the taste quality of rice [21] After weighing 30 g of rice, it was washed with water for 20 s and placed on a stainless-steel pot to ensure that the ratio of rice to water was 1:1.4 or 1.1.35 (Indica rice was 1:1.4, Japonica rice was 1:1.35), then soaked for 30 min, steamed in a rice cooker for 40 min, kept warm for 10 min, and finally placed at room temperature for 1.5 h to determine the taste value of the rice, including the appearance, hardness, stickiness, and taste of the rice. A high taste value often indicated a better cooking and eating quality, including a cooked rice exterior score and taste score.

#### 2.3.4. Data Analysis

The SPSS 20.0 software (Chicago, IL, USA) and Origin 2021 (Northampton, MA, USA) were used for the analysis of variance (ANOVA), mapping, and correlation analysis. Significant differences were deemed to occur at *p* < 0.05. For principal component analysis (PCA), the quality properties of different rice were normalized, the characteristic value and contribution rate were determined, and the quality score of rice was formed.

## 3. Results

### 3.1. Appearance Quality

Appearance quality is the external performance of rice. The appearance of high-quality rice tends to be low in chalkiness. The chalkiness is the white opaque part of the rice [22]. In terms of appearance quality, the length of rice varied from 4.83 mm to 6.34 mm, with an average value of 5.50 mm (Table 1). The detailed data of rice quality was in Appendix A. The width of rice varied from 1.69 to 2.35 mm, with an average value of 2.05 mm. The coefficient of variation between the two was 7.04% and 10.96%. The length and width of rice were mainly determined by the genotype of rice [23], so the length and width of rice did not very much. The chalkiness of rice ranged from 0.81–8.34%, and the coefficient of variation was 44.62%, indicating that the chalkiness was easy to change, and was easily affected by the high temperature during the growth period, resulting in starch and gaps between the particles, and eventually chalkiness [24]. The seven varieties in the experiment may cause chalky changes due to environmental factors.

250N-Yongyou7850 had the highest chalkiness when the length and width of the rice were 4.94 mm and 2.31 mm, respectively. The length of 100N-Tianyuanxiangjing was the highest, reaching 6.34 mm. At this time, the width of the rice was 2.13 mm and the chalkiness was 1.04%. 150N-Yongyou7850 had a maximum width of 2.35, a length of 5.03, and a chalkiness of 6.58%.

### 3.2. Milling Quality

In the milling quality, brown rice percentage, milled rice percentage, and the head rice percentage did not change much. The brown rice percentage varied from 76.09% to 83.85%, the average was 80.05%, and the coefficient of variation was 2.28% (Table 1). The range of milled rice percentage was 64.05% to 73.93%, the average was 69.70%, and the coefficient of variation was 2.28%. The range of the head rice percentage was 59.82% to 72%, the average value was 65.97%, and the coefficient of variation was 4.27%. The coefficient of variation of the head rice percentage was about twice that of the brown rice percentage and the head rice percentage, indicating that for rice under the nitrogen treatment conditions, the head rice percentage is easy to change. It may be that the brown rice percentage and the milled rice percentage often referred to the weight ratio of the rice after milling, and the calculation of the head rice percentage only included the head rice.

In terms of milling quality, the brown rice percentage of 200N-Yongyou4949 reached 83.85%. At this time, the milled rice percentage of rice was 73.93%, the milled rice percentage reached the maximum, and the head rice percentage was 71.61%. 125N-Yongyou4949 had the highest head rice percentage, reaching 72%. At this time, the brown rice percentage of rice was 83.05%, and the milled rice percentage was 73.03%.

### 3.3. Cooking and Eating Quality

In the cooking and eating quality, the change of the RVA coefficient indicated the gelatinization characteristics of rice starch, which referred to the viscosity change of the starch from water absorption to expansion to rupture, which can be laterally reflected in the cooking- and eating quality of rice [25]. The peak viscosity, hold viscosity, and final viscosity of rice were all between 10–20%, while the range of peak time and pasting temperature varies slightly, and the coefficient of variation was 3.79% and 5.41%, respectively, indicating the viscosity change was larger (Table 1). Protein and amylose contents were the two most important factors affecting the cooking and eating quality of rice [26]. The protein content varied from 4.96% to 11.86%, with an average value of 8.41% and a coefficient of variation of 17.89%. The amylose content varied from 14.48% to 23.61%, with an average value of 18.91% and a coefficient of variation of 8.88%. The change range of protein content was more than twice the change range of amylose content, indicating that protein content is more easily affected under nitrogen fertilizer treatment.

The water uptake ratio of rice reflects the degree of water absorption during the cooking process. Rice with a higher water uptake ratio tended to have better cooking- and eating quality [27]. The water uptake ratio of rice varied from 2.46 to 4.70, with an average value of 3.55 and a coefficient of variation of 11.02%. The exterior of cooked rice varied from 6 to 9.40, with an average of 7.33 and a coefficient of variation of 9.34. From the perspective of taste, rice varied from 6.1 to 9.6, with an average value of 7.38 and a coefficient of variation of 9.12. In the taste value of rice, the lowest score was 63 and the highest score was 89. The average value of rice was 75.24, and the coefficient of variation was 9.06%.

0N-Huanghuazhan had the highest peak viscosity, reaching 418.67RVU, protein content of 5.97%, amylose content of 23.61%, and amylose content reached the maximum, water uptake ratio of 4.68, and taste value of 88. The protein content of 0N-Yongyou7850 was the lowest value, reaching 4.96%. At this time, the peak viscosity of the rice was 359.86RVU, the amylose content was 20.28%, the water uptake ratio was 4.02, and the taste value was 81. The water uptake ratio of 175N-Huanghuazhan was 4.70, reaching the maximum, the peak viscosity of rice was 365.83, the protein content was 8.05, and the taste value was 82.

### 3.4. Cluster Analysis

Cluster analysis is to classify objects with the same attributes into one category so that the attributes of different categories can be better predicted [28]. First, the quality of 84 different types of rice was standardized to remove the influence of units, and then cluster analysis was performed based on the inter-group connection and Pearson correlation, and the Euclidean distance was 20 to divide them into four groups (Figure 1).

The group I included 23 types of rice, including Tianyuanxiangjing and Jiafengyou II. In these two varieties, 25 N, 50 N, 75 N, 100 N, 125 N, 150 N, 175 N, 200 N, 250 N, 300 N, and 350 N all appeared twice, 0 N treatment appeared once. The main feature was higher length, width, and peak time (Figure 2).

The group II included 20 types of rice, including two rice varieties, Yongyou7850, and Yongyou4949. In these two varieties, 25 N and 50 N both appeared once, 75 N, 100 N, 125 N, 150 N, 175 N, 200 N, 250 N, 300 N, and 350 N all appeared twice. The main feature was high brown rice percentage, milled rice percentage, and head rice percentage (Figure 3).

The group III includes 16 types of rice, including Meixiangzhan II, Lvyinzhan, and Huanghuazhan. The 75 N and 125 N treatments appeared twice, and the 200 N, 250 N, 300 N, and 350 N treatments appeared three times. The main characteristics are high protein content and chalkiness (Figure 4).

The group IV included 25 types of rice, including Yongyou 7850, Yongyou 4949, Meixiangzhan II, Lvyinzhan, Huanghuazhan, and Jiafengyou II, a total of six rice varieties, of which 0 N treatment appeared six times, 25 N and 50 N treatment appeared four occurrences, one occurrence of 75 N treatment, three occurrences of 100 N treatment, one occurrence of 125 N treatment, three occurrences of 150 N treatment, and three occurrences of 175 N treatment. The main characteristics were high peak viscosity, hold viscosity, final viscosity, and taste value (Figure 4).

The four groups had similar characteristics. It was worth noting that the group IV had the highest frequency of low nitrogen, and the frequency of 0 N had reached 24%, showing higher starch gelatinization viscosity and higher taste value. The group III of high nitrogen occurred more frequently, and the final protein content and chalkiness were higher. The results of cluster analysis showed that the milling quality of group II was higher, the cooking and eating quality of group IV was higher, and the appearance quality of group III was lower.

### 3.5. Principal Component Analysis (PCA)

PCA can convert multiple indicators into multiple comprehensive indicators and determine their relative importance, and ensure that the amount of information after dimensionality reduction is maintained at a sufficiently high level [29]. PCA analysis showed that the KMO value was 0.66 and the significance level was 0.000, indicating that principal component analysis could be performed. In the PCA analysis, the first four main components accounted for 83.342% of the data difference, and their contribution rates were 37.792%, 18.844%, 15.290%, and 11.416% respectively (Table 2). The highest characteristic value of PC1 was 6.425. PC1 was highly correlated with the taste value, followed by water uptake ratio, peak viscosity, the exterior of cooked rice, and amylose content. The brown rice percentage, milled rice percentage and protein content were negative values, indicating that the higher the taste value, the lower the brown rice percentage, milled rice percentage and protein content. The characteristic value of PC2 was 3.203, and the taste and the head rice percentage correspond to the highest eigenvector, which mainly reflected the taste and the head rice percentage. The characteristic value of PC3 was 2.599, and the final viscosity and hold viscosity corresponded to the highest characteristic vector, which mainly reflected the influence of starch gelatinization properties. The characteristic value of PC4 was 1.941, and the pasting temperature and peak time correspond to the highest eigenvector, which also reflected the characteristics of starch gelatinization. Then the four main components were summarized as taste value, taste, final viscosity, and pasting temperature.

### 3.6. Comprehensive Evaluation of Rice Quality

We calculated the principal component model by taking the ratio of the four principal components to the eigenvalues. The sum of the eigenvalues of all principal components were as follows:Y_n_ = PC_n1_ * X1 + PC_n2_ * X2 + PC_n3_ * X3 … + PC_n17_ * X17(1)
F = Y_1_ * 0.4535 + Y_2_ * 0.2261 + Y_3_ * 0.1835 + Y_4_ * 0.1370(2)

Among them, PC_n_ represented the feature vector of the corresponding matrix; Xn represented the standardized index of rice quality; Y_n_ represented the score of each main component in rice; F represented the comprehensive score of rice quality.

From the perspective of comprehensive scores, the score types with a comprehensive score greater than 1 were selected, and a total of 17 quality types were found, with the highest ranking being 0N-Meixiangzhan II (Appendix A). There appeared five times for 0 N treatment, four times for 25 N treatment, four times for 50 N treatment, one time for 75 N treatment, two times for 100 N treatment, and one time for 175 N treatment. In terms of rice varieties, Huanghuazhan and Lvyinzhan each appeared five times, Meixiangzhan II appeared three times, Yongyou 7850 and Yongyou 4949 appeared two times. These types of rice were characterized by lower protein content, higher peak viscosity, and taste value. The comprehensive quality of rice was easily improved under low nitrogen conditions, and Huanghuazhan and Lvyinzhan were easier to obtain better comprehensive quality during the cultivation process.

## 4. Discussion

Rice quality is a complex indicator, which is easily affected by the external environment and genotype. Chalkiness is one of the most important appearance qualities. Chalkiness was most easily affected by temperature. Chalkiness was positively correlated with the temperature at the maturity stage: the temperature at the second week after heading had the closest correlation with the chalkiness [30]. The chalkiness at the maturity stage had the highest correlation with the daily average temperature at the maturity stage of rice. Different rice varieties showed ecological adaptability in different ecological regions [31]. Further research on the chalkiness of rice found that the generation of chalkiness is related to the carbon and nitrogen metabolism of rice, and the decrease in the activity of pyruvate orthophosphate dikinase is related to the chalkiness of rice [32]. The activity of protein-related synthase such as glutamine synthetase and glutamate synthase in rice 15 days after anthesis was correlated with rice chalkiness [33]. The chalkiness of the seven rice varieties we used showed the greatest changes. On the one hand, the growth period of different rice varieties was different, so the temperature during the filling period was different. On the other hand, due to the influence of nitrogen fertilizer, nitrogen fertilizer affects the carbon and nitrogen metabolism in rice grains. Nitrogen metabolism can also cause chalky rice. The chalkiness of Yongyou 7850 changed the most. With the increase of nitrogen fertilizer, it changed from 2.75% to 8.34%, indicating that it is a variety that is easily affected by the environment. The length and width of rice were mainly determined by the genotype of rice varieties, and the external environmental conditions have little effect on the length and width of rice. Previous studies have shown that the effect of nitrogen fertilizer on rice grain type is not significant [34]. Some studies also have shown that higher nitrogen fertilizer reduces the chalkiness and length-to-width ratio of rice [35]. The area of endosperm cells and starch accumulation capacity affected the length and width of rice [36]. In our research, nitrogen fertilizer seemed to increase the length and width of rice, but too much nitrogen fertilizer reduced the width and length of rice, indicating that moderate nitrogen fertilizer promoted the growth of rice endosperm cells, but excessive nitrogen fertilizer inhibited the growth of rice endosperm cells.

The head rice percentage is one of the important processing qualities. The head rice percentage of rice is related to the flowering time of rice, and rice varieties with a shorter flowering time have a higher head rice rate [37]. Nitrogen fertilizer can increase the heading rate of rice [37]. In our research, we found that Yongyou 4949 had the highest rate of whole rice, which was related to the grain type. It has a shorter grain length and does not break easily during processing.

Cooking and eating quality are the core of rice quality. The protein and starch in rice are the first and second chemical components in rice grains. Protein and starch are the most important factors affecting the cooking and eating quality of rice, and the structure of amylopectin also affects the taste of rice [38]. The cooking and eating quality of rice is related to the degree of rice milling [39]. In our research, the milling conditions of all varieties were the same. Under nitrogen fertilizer treatment, a decrease in the cooking and eating quality of rice is often due to an increase in protein content [40]. However, the protein accumulation capacity of different rice varieties is different, so under the same nitrogen treatment conditions, rice shows different taste qualities [41]. In our research, nitrogen fertilizer had little effect on the taste value of Tianyuanxiangjing, indicating that the cooking and eating quality of rice variety was relatively stable, and further research should be conducted on it.

During the heating process of rice, starch gradually begins to gelatinize, which leads to the leaching of amylose. The viscosity of rice increases and then reaches a peak. During the cooling process, as amylose precipitates and starch gel forms, starch began to regenerate [42]. The RVA characteristic curve was often related to the eating quality of rice [43] and the degree of retrogradation of gelatinized rice starch [44]. In our research, we found that Huanghuazhan had the highest peak viscosity under low nitrogen conditions, indicating that Huanghuazhan has the highest gelatinization ability, followed by Lvyinzhan. Nitrogen fertilizer reduces the peak viscosity of all rice varieties, which may be related to the protein content. Protein was a physical barrier around starch, thereby inhibiting starch gelatinization [45].

## 5. Conclusions

Environmental conditions and rice varieties have very complex effects on rice quality. To evaluate rice quality, it is necessary to integrate the appearance, processing, and taste of rice. We evaluated changes in seventeen qualities of seven rice varieties under twelve nitrogen fertilizer treatments and found that seventeen quality traits have different coefficients of variation. Among them, the coefficient of variation of chalkiness and protein content was the largest, 44.60% and 17.89%, respectively. This indicated that chalkiness and protein content were most susceptible to nitrogen fertilizers and rice varieties. Cluster analysis divides them into four categories with different qualities. Each category had its own unique quality characteristics Principal component analysis shows the comprehensive evaluation scores of different rice varieties under different nitrogen fertilizer treatments. The comprehensive quality of rice under low nitrogen conditions is better. Huanghuazhan and Lvyinzhan were easier to obtain better comprehensive quality during the cultivation process. The focus of future rice research should start with reducing protein content and increasing peak viscosity, to improve the quality of rice.

## Figures and Tables

**Figure 1 foods-10-02883-f001:**
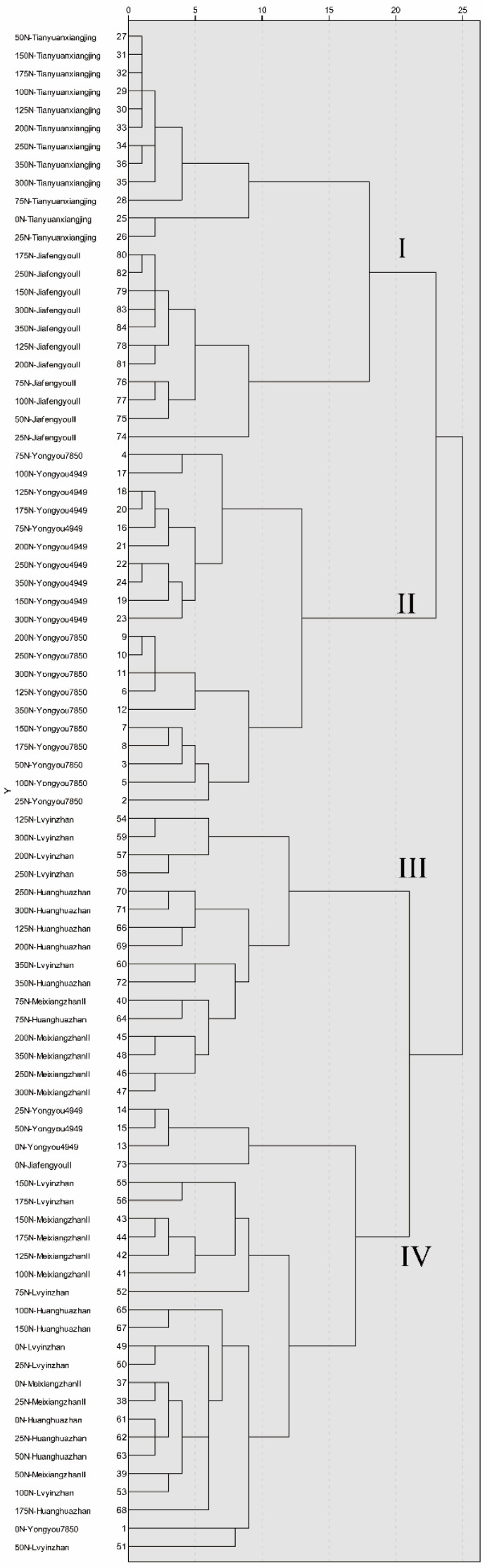
Cluster analysis of rice quality.

**Figure 2 foods-10-02883-f002:**
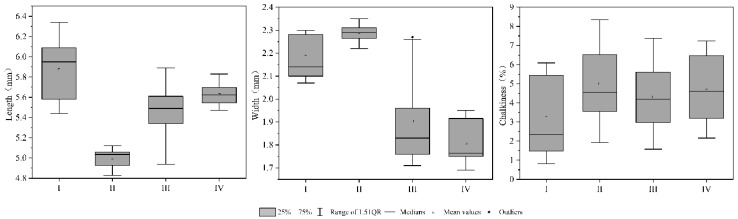
Appearance quality of rice.

**Figure 3 foods-10-02883-f003:**
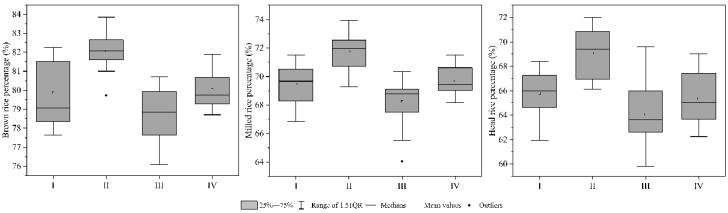
Milling quality of rice.

**Figure 4 foods-10-02883-f004:**
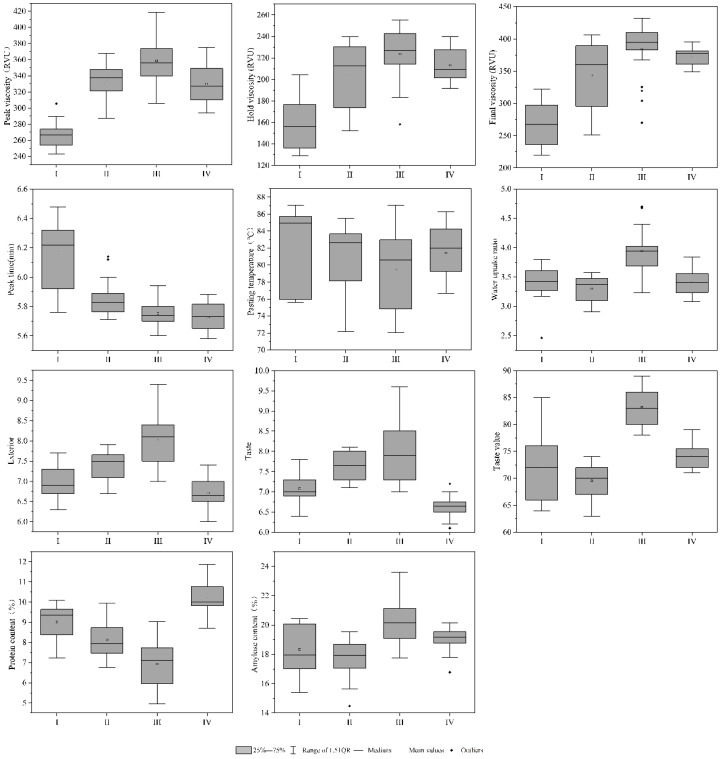
Cooking and eating quality of rice.

**Table 1 foods-10-02883-t001:** Descriptive statistics of rice quality changes.

Parameters	Range	Min	Max	Mean	SD	CV	Nitrogen Fertilizer	Variety
Length (mm)	1.51	4.83	6.34	5.50	0.39	7.04%		**
Width (mm)	0.66	1.69	2.35	2.05	0.22	10.96%		**
Chalkiness (%)	7.53	0.81	8.34	4.27	1.91	44.62%		**
Brown rice percentage (%)	7.76	76.09	83.85	80.05	1.82	2.28%	**	**
Milled rice percentage (%)	9.88	64.05	73.93	69.70	1.82	2.61%	**	**
Head rice percentage (%)	12.18	59.82	72.00	65.97	2.82	4.27%		**
Peak viscosity (RVU)	175.81	242.86	418.67	321.95	43.42	13.49%		**
Hold viscosity (RVU)	126.31	129.06	255.36	199.46	34.91	17.50%		**
Final viscosity (RVU)	211.69	220.14	431.83	340.12	60.58	17.81%		**
Peak time(min)	0.90	5.58	6.48	5.88	0.22	3.79%		**
Pasting temperature (°C)	15.01	72.07	87.07	80.61	4.36	5.41%		**
Protein content (%)	6.90	4.96	11.86	8.41	1.50	17.89%	**	**
Amylose content (%)	9.13	14.48	23.61	18.91	1.68	8.88%		**
Water uptake ratio	2.24	2.46	4.70	3.55	0.39	11.02%	**	**
Exterior	3.40	6.00	9.40	7.33	0.69	9.34%	**	
Taste	3.50	6.10	9.60	7.38	0.67	9.12%	**	
Taste value	26.00	63.00	89.00	75.24	6.82	9.06%	**	**

Range, difference between maximum and minimum; Min, minimum; Max, maxi-mum; Mean, mean of all samples; SD, standard deviation of all samples; CV, coefficient of variation for all samples; ** denote significant differences at the 0.01.

**Table 2 foods-10-02883-t002:** Eigenvectors of corresponding matrices for rice quality traits.

Parameters	PC1	PC2	PC3	PC4
Length (mm)	−0.070	−0.422	−0.106	0.248
Width (mm)	−0.203	0.327	−0.237	−0.055
Chalkiness (%)	0.057	0.055	0.274	−0.459
Brown rice percentage (%)	−0.283	0.269	0.238	0.079
Milled rice percentage (%)	−0.274	0.300	0.185	−0.013
Head rice percentage (%)	−0.238	0.379	0.040	−0.079
Peak viscosity (RVU)	0.290	0.202	0.262	0.108
Hold viscosity (RVU)	0.250	0.069	0.399	0.270
Final viscosity (RVU)	0.256	0.022	0.436	0.178
Peak time (min)	−0.223	−0.083	−0.278	0.404
Pasting temperature (°C)	−0.084	−0.151	0.279	0.429
Protein content (%)	−0.261	−0.287	0.209	−0.025
Amylose content (%)	0.257	−0.138	−0.007	−0.354
Water uptake ratio	0.323	−0.033	−0.163	0.066
Exterior	0.271	0.287	−0.207	0.217
Taste	0.196	0.380	−0.213	0.251
Taste value	0.346	−0.033	−0.182	−0.090

## Data Availability

Data are contained within the article.

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
