# Peer review of "Comprehensive Evaluation of 17 Qualities of 84 Types of Rice Based on Principal Component Analysis"

_foods, 2021, doi:10.3390/foods10112883_

Round 1

Reviewer 1 Report

This paper has examined the properties of rice related to rice quality. The strength of the paper is sampling, data analysis, and results. However, I have concerned for few areas which are mentioned in the attached document. 

Comments for authors

Line54-55: Which varieties belong to indica or japonica groups?

Line 58: Which season and months is considered as rice growing season in China?

Line 114: Range means a difference between the smallest and largest value. What does range stand for? Please clarify. What do SD and CV stand for. Mention an elaboration form of SD and CV. Have you looked at whether these coefficient of variations are significant or not between samples?  You need to add a column with a significant value for coefficient of variation.

Line 188-190. This sentence is long and confusing. Revised of this sentence is required.

Line 267: Grain appearance is evaluated by its size and shape, length and width, length, and width percentage (L/W), translucency, and chalkiness of the endosperm. In the discussion, you have mentioned only about grain chalkiness though you have presented results for grain size. It is recommended to discuss every aspects of grain appearance in the discussion. The discussion should capture more properties of rice.

Author Response

Line54-55: Which varieties belong to indica or japonica groups?

Response: Thanks! We have described the rice varieties in more detail, and the added information is as follows:

Indica rice included Huanghuazhan, Meixiangzhan, Lvyinzhan and Jiafengyouâ…¡. Japonica rice included Yongyou7850, Yongyou4949 and Tianyuanxiangjing.

Line 58: Which season and months is considered as rice growing season in China?

Response: Thanks! The growing period of rice in China is from May(spring) to November(autumn). The rice varieties we used in the article were planted from June (summer) to November (autumn).

Line 114: Range means a difference between the smallest and largest value. What does range stand for? Please clarify. What do SD and CV stand for. Mention an elaboration form of SD and CV. Have you looked at whether these coefficient of variations are significant or not between samples?  You need to add a column with a significant value for coefficient of variation.

Response: Thanks! The range represents the difference between the maximum and minimum values of each indicator in all samples. SD and CV represent the standard deviation and the coefficient of variation, respectively. These are descriptive statistics for 17 indicators in 84 samples. I will add a column with a significant value for coefficient of variation. The specific table information is as follows:

Table 1. Descriptive statistics of rice quality changes.

Parameters

Range

Min

Max

Mean

SD

CV

Nitrogen fertilizer

Variety

Length(mm)

1.51

4.83

6.34

5.50

0.39

7.04%

**

Width(mm)

0.66

1.69

2.35

2.05

0.22

10.96%

**

Chalkiness(%)

7.53

0.81

8.34

4.27

1.91

44.62%

**

Brown rice percentage (%)

7.76

76.09

83.85

80.05

1.82

2.28%

**

**

Milled rice percentage (%)

9.88

64.05

73.93

69.70

1.82

2.61%

**

**

Head rice percentage (%)

12.18

59.82

72.00

65.97

2.82

4.27%

**

Peak viscosity(RVU)

175.81

242.86

418.67

321.95

43.42

13.49%

**

Hold viscosity (RVU)

126.31

129.06

255.36

199.46

34.91

17.50%

**

Final viscosity (RVU)

211.69

220.14

431.83

340.12

60.58

17.81%

**

Peak time(min)

0.90

5.58

6.48

5.88

0.22

3.79%

**

Pasting temperature(℃)

15.01

72.07

87.07

80.61

4.36

5.41%

**

Protein content(%)

6.90

4.96

11.86

8.41

1.50

17.89%

**

**

Amylose content(%)

9.13

14.48

23.61

18.91

1.68

8.88%

**

Water uptake ratio

2.24

2.46

4.70

3.55

0.39

11.02%

**

**

Exterior

3.40

6.00

9.40

7.33

0.69

9.34%

**

Taste

3.50

6.10

9.60

7.38

0.67

9.12%

**

Taste value

26.00

63.00

89.00

75.24

6.82

9.06%

**

**

Range, difference between maximum and minimum; Min, minimum; Max, maxi-mum; Mean, mean of all samples; SD, standard deviation of all samples; CV, coefficient of variation for all samples; * and ** denote significant differences at the 0.05 and 0.01, respectively.

Line 188-190. This sentence is long and confusing. Revised of this sentence is required.

Response: Thanks! We have revised specific sentences, and the specific revised information is as follows:

The group â…  included 23 types of rice, including Tianyuanxiangjing and Jiafengyouâ…¡. In these two varieties, 25N ,50N, 75N, 100N, 125N, 150N, 175N, 200N, 250N, 300N, and 350N all appeared twice,0N treatment appeared once. The main feature was higher length, width and peak time (Figure 2)

Line 267: Grain appearance is evaluated by its size and shape, length and width, length, and width percentage (L/W), translucency, and chalkiness of the endosperm. In the discussion, you have mentioned only about grain chalkiness though you have presented results for grain size. It is recommended to discuss every aspects of grain appearance in the discussion. The discussion should capture more properties of rice.

Response: Thanks! we will add the discussion of the more properties of rice to the discussion, the discussion part is revised as follows:

Rice quality was a complex indicator, which was easily affected by the external environment and genotype. Chalkiness was one of the most important appearance qualities. Chalkiness was most easily affected by temperature. Chalkiness is positively correlated with the temperature at the maturity stage, the temperature at the second week after heading has the closest correlation with the chalkiness [30]. The chalkiness at the maturity stage has the highest correlation with the daily average temperature at the maturity stage of rice. Different rice varieties show ecological adaptability in different ecological regions [31]. Further research on the chalkiness of rice found that the generation of chalkiness is related to the carbon and nitrogen metabolism of rice, and the decrease in the activity of pyruvate orthophosphate dikinase is related to the chalkiness of rice[32]. The activity of protein-related synthase such as glutamine synthetase and glutamate synthase in rice 15 days after anthesis is correlated with rice chalkiness [33]. The chalkiness of the 7 rice varieties we used showed the greatest changes. On the one hand, the growth period of different rice varieties was different, so the temperature during the filling period was different. On the other hand, due to the influence of nitrogen fertilizer, nitrogen fertilizer affects the carbon and nitrogen metabolism in rice grains. Nitrogen metabolism can also cause chalky rice. The chalkiness of Yongyou 7850 changed the most. With the increase of nitrogen fertilizer, it changed from 2.75% to 8.34%, indicating that it is a variety that is easily affected by the environment. The length and width of rice were mainly determined by the genotype of rice varieties, and the external environmental conditions have little effect on the length and width of rice. Previous studies have shown that the effect of nitrogen fertilizer on rice grain type was not significant [34] . Some studies also have shown that higher nitrogen fertilizer reduces the chalkiness and length to width ratio of rice[35]. The area of endosperm cells and starch accumulation capacity affected the length and width of rice [36] In our research, nitrogen fertilizer seemed to increase the length and width of rice, but too much nitrogen fertilizer reduces the width and length of rice, indicating that moderate nitrogen fertilizer promoted the growth of rice endosperm cells, but excessive nitrogen fertilizer inhibited the growth of rice endosperm cells.

The head rice percentage is one of the important processing qualities. The head rice percentage of rice is related to the flowering time of rice, and rice varieties with a shorter flowering time have a higher head rice rate [37]. Nitrogen fertilizer can increase the heading rate of rice [37]. In our research, we found that Yongyou 4949 had the highest rate of whole rice, which was related to the grain type. It has a shorter grain length and is not easy to break during processing.

Cooking and eating quality are the core of rice quality. The protein and starch in rice are the first and second chemical components in rice grains. Protein and starch are the most important factors affecting the cooking and eating quality of rice, the structure of amylopectin also affects the taste of rice [38]. The cooking and eating quality of rice is related to the degree of rice milling [39]. In our research, the milling conditions of all varieties were the same. Under nitrogen fertilizer treatment, the decrease in the cooking and eating quality of rice is often due to the increase in protein content [40], However, the protein accumulation capacity of different rice varieties is different, so under the same nitrogen treatment conditions, the rice shows different taste qualities [41]. In our research, nitrogen fertilizer had little effect on the taste value of Tianyuanxiangjing, indicating that the cooking and eating quality of rice variety was relatively stable, and further research should be conducted on it.

During the heating process of rice, starch gradually begins to gelatinize, which leaded to the leaching of amylose. The viscosity of rice increased and then reached the peak viscosity. During the cooling process, as amylose precipitates and starch gel forms, starch began to regenerate [42] . The RVA characteristic curve was often related to the eating quality of rice [43] . In our research, we found that Huanghuazhan had the highest peak viscosity under low nitrogen conditions, indicating that Huanghuazhan has the highest gelatinization ability, followed by Lvyinzhan. Nitrogen fertilizer reduces the peak viscosity of all rice varieties, which may be related to the protein content. Protein was a physical barrier around starch, thereby inhibiting starch gelatinization [44].

Reviewer 2 Report

This manuscript is on the multivariate analysis for the quality evaluation of various rice samples, and its scientific field is suitable for this journal. But, Introduction is too simple and insufficient to explain the scientific background and meaning of this paper. For example, reference 1, 2, 3, 4 are not suitable for the description. Methods are not described accurately for the readers to chase the experiments because official method was not used and published references were not cited. In Results, raw data is not shown, and only the results of statistical treatments, such as PCA and cluster analysis. Therefore, we can not judge the results are right or not, and readers can not get valuable information about the characteristics of rice cultivars and effects of different nitrogen fertilizer application, results of quality evaluation of pasting properties, taste scores, etc. Discussion is too short and more than half was on the chalkiness and description about the quality is not explained enough. The results of pasting properties are ignored in discussion. Authors mentioned final viscosity in Abstract, but, they recommend increasing of peal viscosity in Conclusion. They cited only new references and did not cite important references, such as "Rice" or AACC approved method, etc.         

Author Response

This manuscript is on the multivariate analysis for the quality evaluation of various rice samples, and its scientific field is suitable for this journal. But, Introduction is too simple and insufficient to explain the scientific background and meaning of this paper. For example, reference 1, 2, 3, 4 are not suitable for the description. Methods are not described accurately for the readers to chase the experiments because official method was not used and published references were not cited. In Results, raw data is not shown, and only the results of statistical treatments, such as PCA and cluster analysis. Therefore, we can not judge the results are right or not, and readers can not get valuable information about the characteristics of rice cultivars and effects of different nitrogen fertilizer application, results of quality evaluation of pasting properties, taste scores, etc. Discussion is too short and more than half was on the chalkiness and description about the quality is not explained enough. The results of pasting properties are ignored in discussion. Authors mentioned final viscosity in Abstract, but, they recommend increasing of peal viscosity in Conclusion. They cited only new references and did not cite important references, such as "Rice" or AACC approved method, etc.

Response: Thanks! I will make amendments based on your comments, the specific amendment information is as follows:

Introduction is too simple and insufficient to explain the scientific background and meaning of this paper.

Response: Thanks! The introduction is revised as follows:

Rice is the staple food of two-thirds of the world, and the production of rice is very important [1]. China is the most important rice producer in the world, and about half of the people feed on rice as their staple food [2]. Different from wheat, corn, and other crops, the rice is usually converted into polished rice after being hulled and milled, which can then be eaten by people [3]. With the increase of living standards, people pay more and more attention to the quality of rice [4]. The quality of rice includes the appearance quality, milling quality, cooking and eating quality of rice, etc. Appearance quality refers to the length, width, and chalkiness of rice. In terms of grain type, people in some countries like long rice grains. Long-grain rice has soft grains and has a higher elongation rate after being boiled [5]. Chalkiness affects the overall appearance of rice, and people tend to prefer rice without chalkiness [6]. Milling quality usually refers to the proportion of rice that has been dehulled and milled into polished rice. After the rice with poor processing quality is milled, the proportion of rice milled into polished rice decreases, indicating that there is more waste in the processing process [7]. Cooking and eating quality refers to the taste of rice, high taste rice has a brighter appearance and softer taste [8]. In the past, it was often only possible to taste the rice directly in order to correctly evaluate the taste quality of the rice [9]. However, this method is often cumbersome, and the results are inaccurate due to differences in age, region, and preference among different people. In addition to direct tasting, the eating quality of rice can also be measured indirectly through RVA. High eating quality rice tends to have a higher peak viscosity and a lower gelatinization temperature [10]. A recent rice taste analyzer can evaluate the taste value of rice. Rice with high cooking and eating quality tends to have a higher taste value[11].

Nitrogen fertilizer is one of the most important factors affecting the cooking and eating quality of rice. Appropriate nitrogen increases the yield of rice, but it will affect the quality of rice. Nitrogen fertilizer will improve the milling quality of rice [12], reduce the taste of rice, and ultimately lead to the poor taste of rice [13]. Recently, a compre-hensive evaluation analysis on foxtail millet quality indicated that future research on foxtail millet should focus on increasing protein content and amino acids[14]. Some key indicators can be used to evaluate the high temperature tolerance of different rice varieties, to select the best varieties[15]. However, there are not many researches on rice quality, and research is needed to guide rice production through a comprehensive analysis of rice quality. For a long time, the effect of nitrogen on rice quality was often concentrated in a single quality study. , There are often few rice varieties used, the quality evaluation of rice is not representative, and there are no comprehensive evalu-ations of rice quality.

Therefore, we used principal component analysis to evaluate rice comprehensively quality, it was helpful to help researchers determine good rice varieties and cultivation conditions. In this study, 7 widely grown rice varieties, including 3 japonica rice varie-ties and 4 indica rice varieties, were subjected to 12 nitrogen fertilizer treatments in the field test, resulting in 84 different types of 17 rice qualities. The quality of rice (ap-pearance quality, milling quality, cooking and eating quality) for a comprehensive evaluation, to provide their suggestions for high quality rice production.

Methods are not described accurately for the readers to chase the experiments because official method was not used and published references were not cited.

Response: Thanks! The test method has been carefully described, and a more detailed description and references are revised as follows:

2.1. Appearance quality

A rice appearance quality analyzer (SC-E, Hangzhou Wanshen Test Technology Corporation, China) was used to analyze the appearance quality of milled rice[16], in-cluding length, width and chalkiness. Grains containing more than 20% white belly, white center, white back or a combination of these are considered chalk particles.

Fang, C.; Hu, X.; Sun, C.; Duan, B.; Xie, L.; Zhou, P. Simultaneous Determination of Multi Rice Quality Parameters Using Image Analysis Method. Food Analytical Methods 2015, 8, 70-78, doi:10.1007/s12161-014-9870-2.

2.3.1. Protein and amylose contents

The rice samples were pulverized (Foss, Hilleroed, Denmark) and passed through a 100-mesh aperture. the N concentration of milled rice was determined using an Ele-mental analyzer (Elementar, Langenselbold, Germany), then converted into protein content using a conversion factor (5.95) [18]. The amylose content was determined by the iodometric method [19]. An Epoch Microplate Spectrophotometer (BioTek, Vermont, USA) was used to measure the color at 620 nm. The amylose content values were calculated from a standard curve established using mixture solutions of amylose and amylopectin.

Jones, D.B. Factors for converting percentages of nitrogen in foods and feeds into percentages of proteins; US Department of Agriculture: 1931.

2.3.2. RVA profiles

A rapid viscosity analyzer RVA (Newport, Warri wood, Australia) was used to determine the RVA profiles of rice. According to AACC (American Association of Ce-real Chemists) procedures and Windows Thermal Cycle (TCW3) for data processing and data analysis [20]. About 3g of rice flour is mixed with 25ml of water and then put on the aluminum can. The RVA program first heated at 50°C for one minute, then heated to 95°C in 3.75 minutes, and then heated at 95°C for 2.5 minutes. It was then cooled to 50°C in 3.75 minutes and held for 1.4 minutes. Head rice samples (2 g) were cooked in 20 ml distilled water for 20 min in a boiling water bath. The contents were drained and cooked samples were then weighed accurately and the water uptake ratio was calculated.

AACC, I. Approved Methods of the AACC. Association of Cereal Chemists, St. Paul 2000.

2.3.3. Taste values

The rice taste analyzer (Satake, Hiroshima, Japan) was used to determine the taste value of rice. The rice taste analyzer converted various physical and chemical properties of rice into taste values to determine the taste quality of rice [21]. After weighing 30g of rice, washed it with water within 20 seconds and placed it on a stainless-steel pot to ensure that the ratio of rice to water was 1:1.4 or 1.1.35 (Indica rice was 1:1.4, Japonica rice was 1:1.35), then soaked for 30 minutes, steamed in a rice cooker for 40 minutes and kept warm for 10 minutes, and finally placed at room temperature for 1.5 hours to determine the taste value of the rice, including the appearance, hardness, stickiness, and taste of the rice. A high taste value often indicated a better cooking and eating quality, included cooked rice exterior score and taste score.

Champagne, E.T.; Richard, O.A.; Bett, K.; Grimm, C.; Vinyard, B.; Webb, B.; McClung, A.; Barton, F.; Lyon, B.; Moldenhauer, K. Quality evaluation of US medium-grain rice using a Japanese taste analyzer. Cereal chemistry (USA) 1996.

In Results, raw data is not shown, and only the results of statistical treatments, such as PCA and cluster analysis. Therefore, we can not judge the results are right or not, and readers can not get valuable information about the characteristics of rice cultivars and effects of different nitrogen fertilizer application, results of quality evaluation of pasting properties, taste scores, etc.

Response: Thanks! Since the multivariate analysis only shows the results and cannot reflect the various effects of nitrogen fertilizer on rice quality, I will follow the practices in some articles and add data on the various qualities and paste attributes of the 84 samples in the supplementary documents to ensure the results are true and reliable.

Discussion is too short and more than half was on the chalkiness and description about the quality is not explained enough. The results of pasting properties are ignored in discussion.

Response: Thanks! There will be more detailed discussions next, and the discussion are revised as follows:

Rice quality was a complex indicator, which was easily affected by the external environment and genotype. Chalkiness was one of the most important appearance qualities. Chalkiness was most easily affected by temperature. Chalkiness is positively correlated with the temperature at the maturity stage, the temperature at the second week after heading has the closest correlation with the chalkiness [30]. The chalkiness at the maturity stage has the highest correlation with the daily average temperature at the maturity stage of rice. Different rice varieties show ecological adaptability in different ecological regions [31]. Further research on the chalkiness of rice found that the generation of chalkiness is related to the carbon and nitrogen metabolism of rice, and the decrease in the activity of pyruvate orthophosphate dikinase is related to the chalkiness of rice [32]. The activity of protein related synthase such as glutamine synthetase and glutamate synthase in rice 15 days after anthesis is correlated with rice chalkiness [33]. The chalkiness of the 7 rice varieties we used showed the greatest changes. On the one hand, the growth period of different rice varieties was different, so the temperature during the filling period was different. On the other hand, due to the influence of nitrogen fertilizer, nitrogen fertilizer affects the carbon and nitrogen metabolism in rice grains. Nitrogen metabolism can also cause chalky rice. The chalkiness of Yongyou 7850 changed the most. With the increase of nitrogen fertilizer, it changed from 2.75% to 8.34%, indicating that it is a variety that is easily affected by the environment. The length and width of rice were mainly determined by the genotype of rice varieties, and the external environmental conditions have little effect on the length and width of rice. Previous studies have shown that the effect of nitrogen fertilizer on rice grain type was not significant [34] . Some studies also have shown that higher nitrogen fertilizer reduces the chalkiness and length to width ratio of rice [35]. The area of endosperm cells and starch accumulation capacity affected the length and width of rice [36] In our research, nitrogen fertilizer seemed to increase the length and width of rice, but too much nitrogen fertilizer reduces the width and length of rice, indicating that moderate nitrogen fertilizer promoted the growth of rice endosperm cells, but excessive nitrogen fertilizer inhibited the growth of rice endosperm cells.

The head rice percentage is one of the important processing qualities. The head rice percentage of rice is related to the flowering time of rice, and rice varieties with a shorter flowering time have a higher head rice rate [37]. Nitrogen fertilizer can increase the heading rate of rice [37]. In our research, we found that Yongyou 4949 had the highest rate of whole rice, which was related to the grain type. It has a shorter grain length and is not easy to break during processing.

Cooking and eating quality are the core of rice quality. The protein and starch in rice are the first and second chemical components in rice grains. Protein and starch are the most important factors affecting the cooking and eating quality of rice, the structure of amylopectin also affects the taste of rice [38]. The cooking and eating quality of rice is related to the degree of rice milling [39]. In our research, the milling conditions of all varieties were the same. Under nitrogen fertilizer treatment, the decrease in the cooking and eating quality of rice is often due to the increase in protein content [40], However, the protein accumulation capacity of different rice varieties is different, so under the same nitrogen treatment conditions, the rice shows different taste qualities [41]. In our research, nitrogen fertilizer had little effect on the taste value of Tianyuanxiangjing, indicating that the cooking and eating quality of rice variety was relatively stable, and further research should be conducted on it.

During the heating process of rice, starch gradually begins to gelatinize, which leaded to the leaching of amylose. The viscosity of rice increases and then reaches the peak viscosity. During the cooling process, as amylose precipitates and starch gel forms, starch began to regenerate [42] . The RVA characteristic curve was often related to the eating quality of rice [43] . In our research, we found that Huanghuazhan had the highest peak viscosity under low nitrogen conditions, indicating that Huanghuazhan has the highest gelatinization ability, followed by Lvyinzhan. Nitrogen fertilizer reduces the peak viscosity of all rice varieties, which may be related to the protein content. Protein was a physical barrier around starch, thereby inhibiting starch gelatinization [44].

Authors mentioned final viscosity in Abstract, but, they recommend increasing of peal viscosity in Conclusion.

Response: Thanks! In the abstract, I mentioned the final viscosity, because the coefficient of variation of the final viscosity is large, indicating that the final viscosity is more susceptible to the influence of the external environment and rice varieties. In the RVA curve, the coefficient of variation between the final viscosity and the peak viscosity is relatively close, but in the PCA analysis, the peak viscosity has a greater impact on rice quality, and the peak viscosity can better indicate the change in the comprehensive quality of rice, so I choose the peak viscosity instead of final viscosity. I will carefully modify the abstract and conclusions, ensure that the abstract and conclusion are consistent, the abstract and conclusion is revised as follows:

Abstract: Rice quality is a complex indicator, and people pay more and more attention to the quality of rice. Therefore, we used 7 rice varieties for 12 nitrogen fertilizer treatments and obtained 84 rice types with 17 qualities. It was found that 17 quality traits had different coefficients of variation. Among them, the coefficient of variation of chalkiness and protein content was the largest, 44.60% and 17.89% respectively. The cluster analysis method was used to define 4 categories of different rice qualities. The principal component analysis method was used to comprehensively evaluate 17 qualities of 84 rice. It was found that rice quality was better under low nitrogen conditions, Huanghuazhan and Lvyinzhan were easier to obtain better comprehensive rice quality during cultivation. Future rice research should focus on reducing protein content and increasing peak viscosity.

  1. Conclusion

Environmental conditions and rice varieties have very complex effects on rice quality. To evaluate rice quality, it is necessary to integrate the appearance, processing and taste of rice. We evaluated the changes in 17 qualities of 7 rice varieties under 12 nitrogen fertilizer treatments and found that 17 quality traits have different coefficients of variation. Among them, the coefficient of variation of chalkiness and protein content was the largest, 44.60% and 17.89% respectively. This indicated that chalkiness and protein content were most susceptible to nitrogen fertilizers and rice varieties. Cluster analysis divides them into 4 categories with different qualities. Each category had its own unique quality characteristics. Principal component analysis shows the comprehensive evaluation scores of different rice varieties under different nitrogen fertilizer treatments. The comprehensive quality of rice under low nitrogen conditions is better. Huanghuazhan and Lvyinzhan were easier to obtain better comprehensive quality during the cultivation process. The focus of fu-ture rice research should start with reducing protein content and increasing peak viscosity, to improve the quality of rice.

They cited only new references and did not cite important references, such as "Rice" or AACC approved method, etc.

Response: Thanks! I will carefully modify the methods and references, and the method and references are revised as follows:

2.1. Appearance quality

A rice appearance quality analyzer (SC-E, Hangzhou Wanshen Test Technology Corporation, China) was used to analyze the appearance quality of milled rice [16], in-cluding length, width and chalkiness. Grains containing more than 20% white belly, white center, white back or a combination of these are considered chalk particles.

Fang, C.; Hu, X.; Sun, C.; Duan, B.; Xie, L.; Zhou, P. Simultaneous Determination of Multi Rice Quality Parameters Using Image Analysis Method. Food Analytical Methods 2015, 8, 70-78, doi:10.1007/s12161-014-9870-2.

2.3.1. Protein and amylose contents

The rice samples were pulverized (Foss, Hilleroed, Denmark) and passed through a 100-mesh aperture. the N concentration of milled rice was determined using an Ele-mental analyzer (Elementar, Langenselbold, Germany), then converted into protein content using a conversion factor (5.95) [18]. The amylose content was determined by the iodometric method [19]. An Epoch Microplate Spectrophotometer (BioTek, Vermont, USA) was used to measure the color at 620 nm. The amylose content values were calculated from a standard curve established using mixture solutions of amylose and amylopectin.

Jones, D.B. Factors for converting percentages of nitrogen in foods and feeds into percentages of proteins; US Department of Agriculture: 1931.

2.3.2. RVA profiles

A rapid viscosity analyzer RVA (Newport, Warri wood, Australia) was used to determine the RVA profiles of rice. According to AACC (American Association of Ce-real Chemists) procedures and Windows Thermal Cycle (TCW3) for data processing and data analysis [20]. About 3g of rice flour is mixed with 25ml of water and then put on the aluminum can. The RVA program first heated at 50°C for one minute, then heated to 95°C in 3.75 minutes, and then heated at 95°C for 2.5 minutes. It was then cooled to 50°C in 3.75 minutes and held for 1.4 minutes. Head rice samples (2 g) were cooked in 20 ml distilled water for 20 min in a boiling water bath. The contents were drained and cooked samples were then weighed accurately and the water uptake ratio was calculated.

AACC, I. Approved Methods of the AACC. Association of Cereal Chemists, St. Paul 2000.

2.3.3. Taste values

The rice taste analyzer (Satake, Hiroshima, Japan) was used to determine the taste value of rice. The rice taste analyzer converted various physical and chemical properties of rice into taste values to determine the taste quality of rice [21]. After weighing 30g of rice, washed it with water within 20 seconds and placed it on a stainless-steel pot to ensure that the ratio of rice to water was 1:1.4 or 1.1.35 (Indica rice was 1:1.4, Japonica rice was 1:1.35), then soaked for 30 minutes, steamed in a rice cooker for 40 minutes and kept warm for 10 minutes, and finally placed at room temperature for 1.5 hours to determine the taste value of the rice, including the appearance, hardness, stickiness, and taste of the rice. A high taste value often indicated a better cooking and eating quality, included cooked rice exterior score and taste score.

Champagne, E.T.; Richard, O.A.; Bett, K.; Grimm, C.; Vinyard, B.; Webb, B.; McClung, A.; Barton, F.; Lyon, B.; Moldenhauer, K. Quality evaluation of US medium-grain rice using a Japanese taste analyzer. Cereal chemistry (USA) 1996.

Round 2

Reviewer 2 Report

This manuscript was properly revised by adding "discussion", "supplemental table" for the data of quality evaluations, and "references" according to the comments by the reviewer. Therefore, it could be published in Foods after minor revisions by adding references. 

(1) should be "N.W. Childs: Production and utilization of rice. In Rice-Chemistry and Technology-Ed by E.T. Champagne, pp.1-23, AACC Inc. St Paul. MN, USA, 2004".

or

(1) should be "Xin Wei and Xuehui Huang: Origin, taxonomy, and phylogenetics of rice. In Rice-Chemistry and Technology-Ed by Jinsong Bao, pp.1-29, AACC Intl, St Paul.MN, USA, 2019"

L364: eating quality of rice (43) and the degree of retrogradation of gelatinized rice starch (44). 

(44): S. Nakamura et al.: Foods 2021, 10, 987.

(45): Hamer R.J. 

The present author recommends to move supplemental table on the components and quality of each rice samples to the main table. But it is for the decision by the authors. 

Author Response

This manuscript was properly revised by adding "discussion", "supplemental table" for the data of quality evaluations, and "references" according to the comments by the reviewer. Therefore, it could be published in Foods after minor revisions by adding references.

(1) should be "N.W. Childs: Production and utilization of rice. In Rice-Chemistry and Technology-Ed by E.T. Champagne, pp.1-23, AACC Inc. St Paul. MN, USA, 2004".

or

(1) should be "Xin Wei and Xuehui Huang: Origin, taxonomy, and phylogenetics of rice. In Rice-Chemistry and Technology-Ed by Jinsong Bao, pp.1-29, AACC Intl, St Paul.MN, USA, 2019"

Response: Thanks! We have revised the references, and the revised information is as follows:

Wei, X.; Huang, X. 1 - Origin, taxonomy, and phylogenetics of rice. In Rice (Fourth Edition), Bao, J., Ed.; AACC International Press: 2019; pp. 1-29.

L364: eating quality of rice (43) and the degree of retrogradation of gelatinized rice starch (44).

(44): S. Nakamura et al.: Foods 2021, 10, 987.

(45): Hamer R.J.

Response: Thanks! We have revised the discussion part, and the revised information is as follows:

During the heating process of rice, starch gradually begins to gelatinize, which leaded to the leaching of amylose. The viscosity of rice increases and then reaches the peak viscosity. During the cooling process, as amylose precipitates and starch gel forms, starch began to regenerate [42] . The RVA characteristic curve was often related to the eating quality of rice [43] and the degree of retrogradation of gelatinized rice starch [44]. In our research, we found that Huanghuazhan had the highest peak viscos-ity under low nitrogen conditions, indicating that Huanghuazhan has the highest gelatinization ability, followed by Lvyinzhan. Nitrogen fertilizer reduces the peak viscosity of all rice varieties, which may be related to the protein content. Protein was a physical barrier around starch, thereby inhibiting starch gelatinization [45].

The present author recommends to move supplemental table on the components and quality of each rice samples to the main table. But it is for the decision by the authors.

Response: Thanks! There are too many samples and quality indicators of rice, we decided to include detailed data on rice quality in the supplementary materials.
